

# Do the effects last? A comparison between internal and external focus of attention instructions on golf putting accuracy over multiple days

Miri Nevo[1,2,*], Israel Halperin[1,2,*] and Gal Ziv[3,4]

[1] Department of Health Promotion, School of Public Health, Faculty of Medicine, Tel-Aviv University, Tel Aviv, Israel
[2] Sylvan Adams Sports Institute, Tel Aviv University, Tel Aviv, Israel
[3] Levinsky-Wingate Academic College, Netanya, Israel
[4] Institute of Sport Science and Innovations, Lithuanian Sports University, Kaunas, Lithuania
* These authors contributed equally to this work.

## ABSTRACT

**Background:** The role of attentional focus is a well-explored topic in exercise sciences. Studies generally indicate that external focus (EF) enhances motor performance and learning compared with internal focus (IF). However, most studies only included one or two experimental days which limits participants' exposure to the focus conditions. This raises the question of whether the superiority of EF varies over time.

**Methods:** Accordingly, in this pre-registered within-subject study, we examined the effects of focus instructions on golf-putting performance over four days, with 48–72 h between them. On each day, participants performed 15 putts under three instructional conditions: (1) EF, (2) IF, and (3) control, in a randomized and counterbalanced order.

**Results:** We observed trivial differences in performance between conditions but considerable improvements from day 1 to day 4. When using an exploratory analysis, we found that participants performed better under EF and control conditions compared with the IF condition on day 1, but not on subsequent days.

**Conclusions:** Since IF instructions are more commonly used in practice, we speculate that the two other focus conditions were experienced as more novel, potentially accounting for their superiority on Day 1. Nevertheless, our results question the significance of employing EF to enhance performance.

## INTRODUCTION

The role of attentional focus in motor learning and performance is a widely explored topic within sport and exercise science (*e.g.*, *Chua et al., 2021*; *Nicklas et al., 2022*; *Wulf, 2013*). This area of study examines two main attentional strategies: internal focus (IF) and external focus (EF). IF involves participants focusing on specific body parts or muscle groups during a motor task, such as contracting the arm muscles while throwing.

Corresponding author
Gal Ziv, galziv@yahoo.com

Conversely, EF involves participants focusing on the environmental outcomes of task performance, such as the target during a throw (*Wulf, Höß & Prinz, 1998*). These attentional strategies are often compared in terms of their impact on motor learning and physical performance. Studies on motor learning usually employ between-subject designs, where different groups use a distinct focus strategy to practice a task over one or two sessions, followed by a retention test shortly afterward (*e.g., Chiviacowsky, Wulf & Ávila, 2013*). Performance studies often employ within-subject designs, where all participants perform the task under different focus conditions in a counterbalanced manner over a short period, generally 1 or 2 days (*Abdollahipour et al., 2015*; *Porter, Anton & Wu, 2012*; *Wulf et al., 2007*). While studies have generally shown that EF enhances motor performance (*Wulf et al., 2007*; *Zachry et al., 2005*) and learning (*Chiviacowsky, Wulf & Ávila, 2013*; *Shea & Wulf, 1999*; *Wulf & McNevin, 2003*; *Zachry et al., 2005*), as reviewed by *Chua et al. (2021)* and *Wulf (2013)*, recent inquiries have cast doubt on the extent of these effects (*e.g., Montero, 2010*; *Montero, Toner & Moran, 2018*; *Toner & Moran, 2015*; *Wang et al., 2021*; for a review see: *Nicklas et al., 2022*, and *Pompa et al., 2024* in athletes). *Collins, Carson & Toner (2016)* argued for more ecologically valid studies and rigorous methodological approaches in examining the effects of attentional focus. They emphasized the importance of considering theories of skill acquisition and motor control in these studies.

The prevailing explanation for EF's superiority over IF is the constrained action hypothesis (*Wulf, McNevin & Shea, 2001*; *Wulf, Shea & Park, 2001*). This hypothesis suggests that IF promotes conscious control, which hinders motor output by interfering with automatic processes. Conversely, EF promotes unconscious and reflexive processes, leading to greater automaticity and more efficient movement patterns. Here, we tentatively consider an alternative explanation for these effects: novelty. By novelty, we refer to stimuli that are different, new, and perceived as interesting. A significant methodological issue in previous studies is the limited exposure to the attentional focus strategies. Indeed, both motor learning and performance studies typically involve a total of two sessions. It is possible that the limited exposure to focus conditions may impact the effects of EF over time. Since IF instructions are more commonly used than EF in a range of practices, including sports (*van der Graaff et al., 2018*; *Porter, Wu & Partridge, 2010*) and rehabilitation (*Johnson, Burridge & Demain, 2013*) (see *Yamada, Higgins & Raisbeck, 2022* for a review), participants may be more familiar with IF. This could make EF more novel, which can potentially contribute to its superiority. This is because novel stimuli may promote greater interest, enjoyment, and motivation (*Berlyne, 1970*; *González-Cutre & Sicilia, 2019*; *Lakicevic et al., 2020*; *Sylvester et al., 2014*). However, it remains unclear whether this effect endures as participants become more accustomed to EF.

To determine if the novelty of EF accounts for its advantage over IF, a more extended exposure to the focus conditions is required, spanning several days instead of the usual one or two. This approach could reveal several insightful outcomes. First, if the superiority of EF remains over time, it would imply that it is due to factors other than novelty. Second, if the EF superiority increases over time, it could imply that it either augments the positive effects or that IF augments the negative ones. The latter two potential outcomes lend some

support to the constrained action hypothesis. Third, a diminishing advantage for EF over time would indicate a role of novelty in its initial superiority, challenging the constrained action hypothesis.

Therefore, this study aimed to investigate the prolonged effects of attentional focus instructions on performance. We conducted a within-subject experiment where participants performed a golf-putting task under three attentional focus conditions (EF, IF, and control) over 4 days. We hypothesized that on the first day, EF would lead to better performance compared to IF and the control condition. We also hypothesized that this effect would decrease over the following 3 days, although we could not predict the precise timeline and degree of this decline.

## MATERIALS AND METHODS

### Pre-registration and raw data availability

The study was pre-registered on aspredicted.org (https://aspredicted.org/ym8va.pdf). Analyses that were not pre-registered are reported as "exploratory analyses". The raw dataset is available on OSF (https://osf.io/5ypmd/?view_only=1f5603067527419eb179 015759cd083b).

### Participants

The sample size was calculated using a simulation-based power analysis (*Lakens & Caldwell, 2021*). We based our simulation on mean absolute error values in previous studies. We used an expected SD of 10 cm; effect size: Cohen's d = 0.6; correlation between measurements = 0.7. The selected effect size was based on the study of *Chen et al. (2021)*, who reported an effect size of 0.67 between external and internal focus conditions in a similar golf putting task. The analysis showed that 30 participants would provide 80% statistical power. Therefore, 30 physical education students (18 females) between the ages of 18 and 35 years (mean: 25.4 ± 2.7 years) participated in the study. The study was approved by the ethics committee of the Levinsky-Wingate Academic College (approval # 377). All participants signed an informed consent form prior to participation.

### Motor task

Golf putting was used as the motor task. Participants putted golf balls (42.7 mm diameter) from 2.00 m to a regulation golf hole (10.8 cm diameter) on an artificial green (2.75 × 0.91 m), with the purpose of holing as many putts as possible.

### Procedure

The study was conducted over four days with 48–72 h between each day.

#### Day 1

Participants arrived at the motor behavior laboratory and signed an informed consent form. Then, the researcher (M.N) explained and demonstrated the basic putting technique. Instructions included the correct stance with an emphasis on bent knees, straight back, and the correct pendulum-like motion of the arms and golf club. Participants then performed 10 familiarization putts. Following the familiarization stage, they performed 15 putts under

each of the three attentional focus strategies in counterbalanced order (45 putts in total). For the IF condition, participants were verbally instructed to *"concentrate on the pendulum-like movements of your arms."* For the EF condition, they were instructed to *"concentrate on the pendulum-like motion of the golf club,"* and under the control condition, they were told to *"Make sure you are standing at the correct distance from the golf ball."* The external and internal focus instructions were chosen based on previous studies on this topic (*e.g.*, *Wulf, Lauterbach & Toole, 1999*; *Wulf & Su, 2007*).
The researcher repeated the instructions before the first putt and after the fifth and tenth putt in each of the conditions. At the end of the day, participants were asked: (1) to rate on a scale of 1–10 their compliance with the attentional instructions given to them; (2) to rate, from best to worst, their preferred attentional focus; and (3) to rate, from best to worst, their perceived success in each attentional focus.

### Days 2 and 3

Except for the familiarization trials, these days were similar to Day 1.

### Day 4

This day was similar to days 2 and 3, but at the end, the participants were given a chance to say, in their own words, what information they used to rate their preferred attentional focus and their perceived performance under each attentional focus.

## Data analyses

The two main dependent variables were absolute error (AE)—the distance from where the ball landed to the center of the golf hole and variable error (VE)—a measure of consistency or the distance of all putts from their mean. For this purpose, we used a formula provided by *Hancock, Butler & Fischman (1995)*.

### Pre-registered analyses

Because the residuals appeared to be normally distributed, we used a 2-way ANOVA [Condition (internal/external/control) X Day (1–4)] with repeated measures on both factors to assess differences in absolute error and variable error. Bonferroni *post-hoc* analyses and 95% confidence intervals were used for *post-hoc* testing when necessary. In cases where the *p*-value was over 0.05 but under 0.10, and at the same time, the effect size was moderate or above (Cohen's d $\geq$ 0.5 or $\eta_p^2 \geq 0.06$), we considered this finding as practically relevant and discussed it as such.

### Exploratory analyses

We decided to perform a one-way ANOVA for conditions (control, EF, IF) for each day separately. When significant, we used a Holm *post-hoc* test. We also descriptively presented the number of participants that performed best under each of the attentional focus instructions (control, EF, IF). Finally, we created contingency tables to examine the differences between preferred and perceived performance. For all pre-registered and exploratory analyses, alpha was set at 0.05, and when necessary, a Greenhouse-Geisser correction for sphericity was used. Analyses were conducted in JASP (version 0.18.3) (*JASP Team, 2024*).

## RESULTS

Results that are secondary to the main question of this project are reported in the Supplemental Material: differences in preferred attentional focus instructions between days; differences in perceived best attentional focus between days, comparisons of preferred *versus* perceived best attentional focus on each day.

### AE

A two-way ANOVA (Condition X Day) with repeated measures on both factors did not reveal a Condition effect, $F(2, 58) = 2.35$, $p = 0.10$, $\eta^2_p = 0.08$, and no Day X Condition interaction, $F(6, 174) = 1.16$, $p = 0.33$, $\eta^2_p = 0.04$. However, a Day effect was observed, $F(2.48, 72.03) = 20.27$, $p < 0.01$, $\eta^2_p = 0.41$ (Fig. 1). A Bonferroni *post-hoc* analysis showed that the AE in day 4 ($23.91 \pm 7.78$ cm) was lower than the AE in day 1 ($35.17 \pm 10.24$ cm, $p < 0.01$, Cohen's d = 1.04), and day 2 ($30.45 \pm 9.99$ cm, $p < 0.01$, Cohen's d = 0.61), but did not statistically differ from day 3 ($27.89 \pm 6.58$ cm, $p = 0.052$, Cohen's d = 0.37). There were no statistical differences between days 2 and 3 ($p = 0.53$, Cohen's d = 0.24).

### VE

A two-way ANOVA (Condition X Day) with repeated measures on both factors did not reveal a Condition effect, $F(2, 58) = 0.79$, $p = 0.46$, $\eta^2_p = 0.03$, and no Day X Condition interaction, $F(6, 174) = 0.35$, $p = 0.91$, $\eta^2_p = 0.01$. However, a Day effect was observed, $F(3, 87) = 7.24$, $p < 0.01$, $\eta^2_p = 0.20$ (Fig. 2). A Bonferroni *post-hoc* analysis showed that the AE in day 4 ($28.21 \pm 4.69$ cm) was lower than the AE in day 1 ($31.94 \pm 4.34$ cm, $p < 0.01$, Cohen's d = 0.66), and day 2 ($30.80 \pm 4.09$ cm, $p = 0.02$, Cohen's d = 0.46), but did not differ from day 3 ($29.75 \pm 3.67$ cm, $p = 0.41$, Cohen's d = 0.27). There were no differences between days 2 and 3 ($p > 0.99$, Cohen's d = 0.19).

### Adherence to attentional instructions

A one-way repeated measures ANOVA revealed differences in participants' self-reported adherence to attentional instructions between days, $F(2.35, 68.26) = 5.31$, $p = 0.01$, $\eta^2_p = 0.16$. A Bonferroni *post hoc* analysis revealed that the mean self-reported adherence in Day 4 ($8.10 \pm 1.30$) was higher than in Day 1 ($7.03 \pm 0.93$; $p < 0.01$, Cohen's d = 0.83). There were no other differences between days (Day 2: $7.60 \pm 1.43$; Day 3: $7.60 \pm 1.45$).

### Exploratory analyses

***Differences in AE on separate days.*** A one-way ANOVA revealed differences between conditions on Day 1, $F(2, 58) = 5.44$, $p = 0.01$, $\eta^2_p = 0.16$. A Holm *post-hoc* analysis showed a significant difference between IF ($38.48 \pm 10.08$ cm) and control ($32.84 \pm 11.71$ cm, $p = 0.01$, Cohen's d = 0.48), and between IF and EF ($34.19 \pm 13.09$ cm, $p = 0.04$, Cohen's d = 0.37). There were no differences between conditions on day 2, $F(2, 58) = 0.57$, $p = 0.57$, $\eta^2_p = 0.02$; day 3, $F(2, 58) = 0.57$, $p = 0.57$, $\eta^2_p = 0.02$; or day 4, $F(2, 58) = 0.57$, $p = 0.57$, $\eta^2_p = 0.02$.

  ***Differences in VE on separate days.*** A one-way ANOVA revealed no differences in any of the days: Day 1, $F(2, 58) = 0.39$, $p = 0.68$, $\eta^2_p = 0.01$; Day 2, $F(2, 58) = 0.27$, $p = 0.76$,

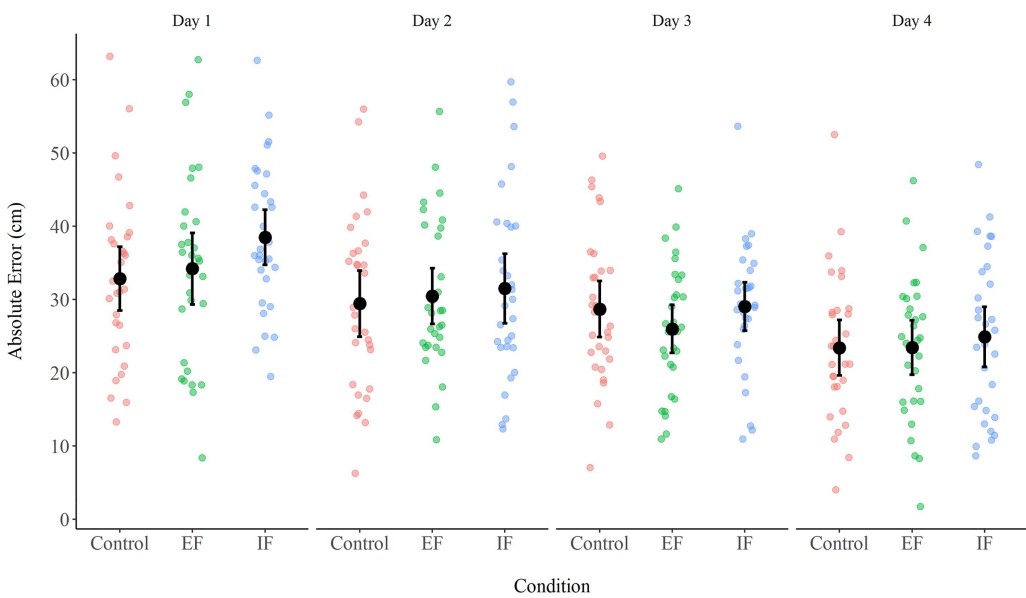

**Figure 1 AE in the three conditions (control, EF, IF) across the 4 days of practice.** Error bars represent 95% CI. (EF = External Focus, IF = Internal Focus).

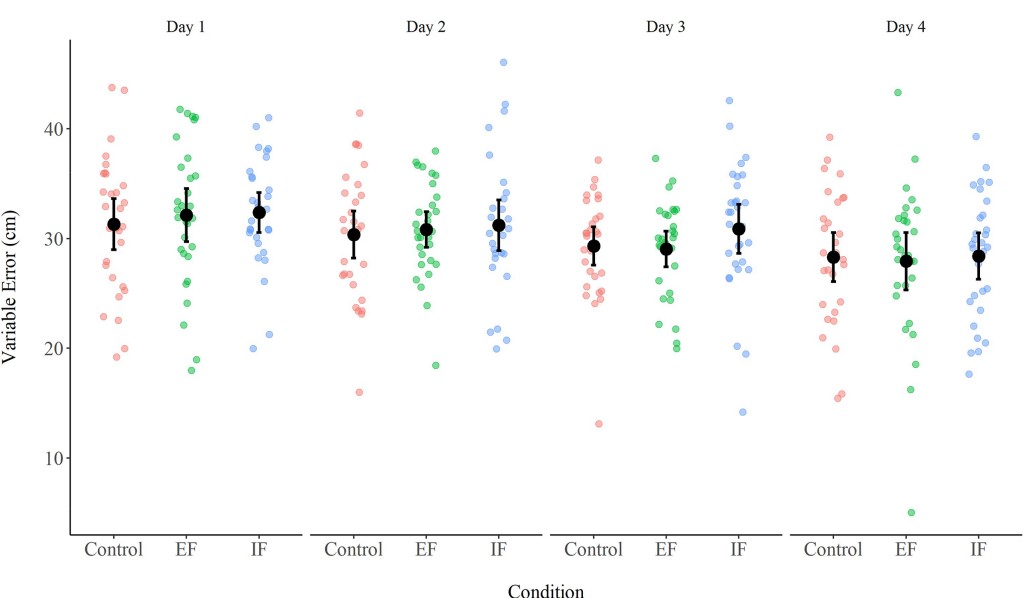

**Figure 2 VE in the three conditions (control, EF, IF) across the 4 days of practice.** Error bars represent 95% CI. (EF = External Focus, IF = Internal Focus).

$\eta^2_p = 0.01$; Day 3, $F(2, 58) = 1.60$, $p = 0.21$, $\eta^2_p = 0.05$; and Day 4, $F(2, 58) = 0.07$, $p = 0.93$, $\eta^2_p = 0.00$.

**Best attentional condition on separate days.** Table 1 presents a count of the attentional conditions that led to the best performance (lowest AE) in all participants for each day. On Day 1, the best performances were achieved under control or EF conditions. On Day 2,

Table 1 The number of participants who achieved their best performance under each of the attentional conditions for each day of practice.

|  | Day 1 | Day 2 | Day 3 | Day 4 |
|---|---|---|---|---|
| Control | 14 | 12 | 10 | 11 |
| External focus | 12 | 7 | 11 | 9 |
| Internal focus | 4 | 11 | 9 | 10 |

the smallest number of participants achieved the best performance under EF. On days 3 and 4, the best performance was divided almost equally between the three conditions.

## DISCUSSION

The present study investigated the impact of EF and IF and control conditions on golf-putting performance over four days. We hypothesized that EF would result in superior putting performance compared to IF and control conditions on the first day, but that this effect would diminish over the following days. Our hypothesis was partially supported. Our pre-registered analysis revealed no discernible differences between conditions. Yet, a separate day-by-day exploratory analysis revealed that on the first day, EF led to superior putting performance (lower AE) compared to IF, although not compared to the control condition. Regardless of the analysis, the advantage of EF did not manifest on other days of practice.

Our findings mostly align with the attentional focus literature. The superior performance with EF compared to IF we observed on the first day is aligned with the results of two systematic reviews and meta-analyses inspecting the immediate impact of focus conditions on task performance (*Chua et al., 2021*; *Nicklas et al., 2022*). The lack of studies comparing focus conditions over an extended duration, as we have done, makes it challenging to compare these results with existing research. However, the observed time effect strongly aligns with evidence linking ongoing practice with improved performance (*e.g.*, *Newell, 1991*; *Newell & Liu, 2012*). Indeed, as *Lee & Wishart (2005)* suggested, *"Although many variables influence how skill improves, no single variable is a stronger predictor of motor skill learning in adults than the amount of practice."* (p. 67). The observed improvement over the days was disassociated from the focus conditions, as no statistical nor practical differences were observed between conditions. Collectively, these findings suggest that if EF has any performance benefits, they are short-lived.

The observation that the EF's positive effects on performance are transient, if they exist at all, may be attributed to the reduced novelty of EF. Since IF is more commonly used in practice (*e.g.*, *Yamada, Higgins & Raisbeck, 2022*), EF might be perceived as more novel, leading to its initially greater effectiveness over IF, as seen in this study. Among other reasons, this is because novel stimuli may promote greater interest and motivation (*González-Cutre & Sicilia, 2019*). The diminished positive effects of EF over time challenge the constrained action hypothesis, which cannot adequately explain these findings.

We note that the novelty explanation in this study is partly challenged by the fact that performance under the control instructions was similar to EF. We speculate that since IF tends to be implemented most frequently in practice (*e.g.*, *Yamada, Higgins & Raisbeck, 2022*), then any instructions, including both EF and control, are experienced as novel. However, this remains to be determined in future studies.

We acknowledge that novelty is only one of the potential explanations for our observations. In addition to novelty, other factors may also impact the results, including participants' adherence to task instructions, individual preferences, and the precise wording of attentional instructions (*e.g.*, *Montero, Toner & Moran, 2018*; *Nicklas et al., 2022*). One or more of these factors can contribute to performance variability while performing and learning a motor task. To illustrate the latter point, we observed heterogeneous reports by participants concerning their preferred attentional focus and their perceived success under the different attentional focus conditions (see Supplemental File). We thus share the concerns of *Montero, Toner & Moran (2018)* about our ability to eliminate confounding factors in attentional focus studies.

Three methodological aspects of this study warrant discussion. First, we focused exclusively on performance and did not assess motor learning by including a retention and a transfer test, as is commonly done in attentional focus studies. Therefore, our results are confined to performance outcomes. Second, we employed a within-subject design, exposing every participant to all focus conditions each day. For more robust findings, future research should consider a between-subject design, where participants are exposed to only one focus condition over multiple days. This method may reduce the influence of other focus conditions, potentially lessening biases such as confusion or preference for a certain focus condition, which might have occurred in our study. Finally, some of the participants in this study may have had previous knowledge of the effects of attentional focus on performance. Although we did not directly ask participants about their previous knowledge of attentional focus effects on performance, we did inquire about their preferred focus during each practice session. The number of participants who preferred each attentional focus (*i.e.*, external, internal, control) was similar (see the Supplemental File), suggesting that prior knowledge did not introduce any meaningful bias. Moreover, if participants had prior knowledge of the benefits of an external focus, it should have biased the results towards improved performance with an external focus throughout the study. As this did not occur, any potential bias only strengthens our finding that the beneficial effects of an external focus of attention are, at most, short-lived.

## CONCLUSIONS

In conclusion, our findings indicate that EF had a small, short-lived advantage over IF in putting performance on the initial day, but this effect did not persist in the following days. These findings imply that the advantages of EF are transient and underscore the potential role of novelty in the superiority of EF over IF. Irrespective of the underlying causes, our results question the significance of using EF instructions to enhance motor task performance.

### Funding

The authors received no funding for this work.

### Competing Interests

The authors declare that they have no competing interests.

### Author Contributions

- Miri Nevo conceived and designed the experiments, performed the experiments, analyzed the data, authored or reviewed drafts of the article, and approved the final draft.
- Israel Halperin conceived and designed the experiments, analyzed the data, authored or reviewed drafts of the article, and approved the final draft.
- Gal Ziv conceived and designed the experiments, analyzed the data, prepared figures and/or tables, authored or reviewed drafts of the article, and approved the final draft.

### Human Ethics

The following information was supplied relating to ethical approvals (*i.e.*, approving body and any reference numbers):

The ethics committee of the Levinsky-Wingate Academic College—Wingate Campus.

### Data Availability

The raw data is available at OSF: Ziv, Gal. 2024. "Continuous Effect of Attentional Focus." OSF. May 28. DOI 10.17605/OSF.IO/5YPMD.

### Supplemental Information

Supplemental information for this article can be found online at http://dx.doi.org/10.7717/peerj.17718#supplemental-information.

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
