# Peer review of "Do the effects last? A comparison between internal and external focus of attention instructions on golf putting accuracy over multiple days"

_PeerJ, doi:10.7717/peerj.17718_

## Round 0.1 · original submission · Major Revisions

Both reviewers reported positive things about the paper. There were reviewer comments concerning the potential influence of subject experience on the results, as well as a request to consider a couple of additional references and the potential of these studies to inform and/or relate to the current study. Please address all of the comments.

·

Basic reporting

The literature references are relevant and clearly identify the main trends within this research area. More could be offered however to highlight the specific concerns by other authors (e.g., Monetro, Toner, Wang etc.) regarding this area of study.

See also: Collins, D., Carson, H.J., & Toner, J. (2016). Letter to the editor concerning the article “Performance of gymnastics skill benefits from an external focus of attention” by Abdollahipour, Wulf, Psotta, & Nieto (2015). Journal of Sports Sciences, 34(13), 1288–1292.

The structure of the article conforms to the acceptable format and the table and figures are produced to a high standard. Data are available as Open Access through an appropriate online repository. The article represents a single, ‘self-contained’ unit of study.

Experimental design

The experimental design conforms with the stated aims and scope of the journal, reflecting an original research article that has applicability within medical or health sciences.

The research has a clear question to be answered and makes an original identification of the gap within existing research; namely, the factor of focus ‘novelty’ within the task.

The research is conducted ethically and utilises accepted methodological procedures within this field of study. Several components of the method are referenced against previously published research, including the instructions provided by participants and data used to conduct a power analysis.

The methods are generally described in sufficient detail; however, more information is required regarding the participants’ experience. Specifically, what was their experience with the golf task? What attentional focus strategies had participants previously used (e.g., in other sports) in order for you to know that an external focus was novel to them? Since the participants were all physical education students, it might be possible that they have experiences in a sport the use of an external focus.

Validity of the findings

All data have been provided, are statistically sound and controlled.
Conclusions are clear and reflect accurately the research question and data collected.

Additional comments

This study is well written and there is a good rationale to suggest the short-term effect. Data are clearly presented which helps to best understand the findings.

I wish to congratulate the authors on their efforts but suggest just a few revisions to improve the article (see comments above).

·

Basic reporting

Introduction
line 38 .... within sport and exercise science
line 54 .. in particular in athlete's population (Pompa et al 2024; see reference below)
line 80-83 Please explain why you want to anticipate the results in the final part of the introduction. I would suggest to delete this part




REF
Pompa, D., et al (2024). Attentional Focus Effects on Lower-Limb Muscular Strength in Athletes: A Systematic Review. The Journal of Strength & Conditioning Research, 38(2), 419-434.

Experimental design

Procedure

Please explain if the participants have previous experience with the task and the FOA instructions

Validity of the findings

- The idea of ‘novelty’ has also been addressed in golf putting with skilled golfers employing a new aiming strategy (i.e., target focussed aiming), in which case Collins et al. (2023; see reference below) found psychophysiological evidence for a more robust switch from external to internal focus of attention when asked to look at the hole. The explanation provided was that looking at the hole helped to avoid distraction from visual stimuli associated with the ball/club/body setup. In other words, it avoided the opportunity to think about task irrelevant information. This idea could be integrated with those being offered in the Discussion for a more contextualised treatment of current literature on this topic

- The discussion section emphasized that the EF group achieved better results than the IF group on day 1. However, it's important to note that the Control group also performed similarly to the EF group on day 1. The similar performance of the Control group deserves to be highlighted and interpreted in the discussion section.

REF
Collins, R., et al. (2023). Where you look during golf putting makes no difference to skilled golfers (but what you look at might!): An examination of Occipital EEG ɑ-power during target and ball focused aiming. International Journal of Sport and Exercise Psychology, 21(3), 456–472.

---

## Round 0.2 · accepted · Accept

Thank you for responding thoroughly to the reviewers' comments. This paper is ready to move forward to the publication phase.

·

Basic reporting

no comment

Experimental design

no comment

Validity of the findings

the authors have included additional detail within the discussion section to add validity to their findings.

Additional comments

I am happy that the authors have addressed the comments in a satisfactory manner and now believe that the manuscript offers an original contribution within the context of the attentional focus literature.

·

Basic reporting

no comment

Experimental design

no comment

Validity of the findings

no comment

Additional comments

The authors' careful consideration of my feedback resulted in a more refined, impactful manuscript, with notable improvements in clarity, structure, and argumentation.